# The Role of NS1 Protein in the Diagnosis of Flavivirus Infections

**DOI:** 10.3390/v15020572

**Published:** 2023-02-19

**Authors:** Ron Fisher, Yaniv Lustig, Ella H. Sklan, Eli Schwartz

**Affiliations:** 1Department of Otolaryngology/Head & Neck Surgery, Hadassah Hebrew; University Medical Center, Jerusalem 91120, Israel; 2Central Virology Laboratory, Ministry of Health, Sheba Medical Center, Ramat-Gan 52621, Israel; 3Sackler Faculty of Medicine, Tel-Aviv University, Tel Aviv 69978, Israel; 4The Center of Geographic Medicine and Tropical Diseases, Sheba Medical Center, Ramat-Gan 52621, Israel

**Keywords:** dengue virus, Zika virus, West Nile virus, NS1 antibodies, NS1 antigen

## Abstract

Nonstructural protein 1 (NS1) is a glycoprotein among the flavivirus genus. It is found in both membrane-associated and soluble secreted forms, has an essential role in viral replication, and modulates the host immune response. NS1 is secreted from infected cells within hours after viral infection, and thus immunodetection of NS1 can be used for early serum diagnosis of dengue fever infections instead of real-time (RT)-PCR. This method is fast, simple, and affordable, and its availability could provide an easy point-of-care testing solution for developing countries. Early studies show that detecting NS1 in cerebrospinal fluid (CSF) samples is possible and can improve the surveillance of patients with dengue-associated neurological diseases. NS1 can be detected postmortem in tissue specimens. It can also be identified using noninvasive methods in urine, saliva, and dried blood spots, extending the availability and effective detection period. Recently, an enzyme-linked immunosorbent assay (ELISA) assay for detecting antibodies directed against Zika virus NS1 has been developed and used for diagnosing Zika infection. This NS1-based assay was significantly more specific than envelope protein-based assays, suggesting that similar assays might be more specific for other flaviviruses as well. This review summarizes the knowledge on flaviviruses’ NS1′s potential role in antigen and antibody diagnosis.

## 1. The Flavivirus NS1

NS1 is a highly conserved nonstructural glycoprotein (46 kDa) encoded by most flaviviruses [1]. These viruses, transmitted mainly by arthropods, are the causative agent of many severe diseases, including dengue fever, West Nile, yellow fever, Zika, and more [2]. Flaviviruses are small, enveloped viruses containing a positive-sense single-strand RNA genome. This genome encodes a large polyprotein precursor that is further processed, co-translationally and post-translationally, by viral and cellular proteases into three structural proteins (capsid protein, membrane protein and envelop protein) and seven nonstructural proteins (NS1, NS2A, NS2B, NS3, NS4A, NS4B, NS5). The structural proteins build the virion, whereas the nonstructural proteins are involved in viral replication [1,2,3]. NS1 can be found both as a membrane-associated, highly stable dimer and a secreted soluble, lipid-associated hexamer [4,5]. In the infected cell, NS1 is synthesized as a soluble monomer and dimerizes after it is post-translationally modified in the lumen of the endoplasmic reticulum [4]. The membrane-associated NS1 can be found both in cellular compartments and on the cell surface [4]. Secreted NS1 is transported to the cell surface and accumulates extracellularly as a hexamer [6], which also binds the plasma membrane of cells through interactions with specific sulfated glycosaminoglycans [7,8]. Soluble NS1 is present in serum and other body fluids [9,10,11,12]. Although it does not have a known catalytic function, it is clear that NS1 is essential for efficient viral RNA replication [5,13]. NS1 proteins also modulate endothelial permeability in a tissue-specific manner, correlating with disease pathology [14]. It has been shown that the immune response against NS1 might damage endothelial cells due to a cross-reaction of the antibodies and immune complex formation, which can elicit autoantibodies that react with platelets and extracellular matrix proteins [15].

Specific antibodies to membrane-associated NS1 and antibodies against soluble NS1 further enhance the activation of the complement system [8]. The connection between those antibodies and the complement system can result in hemorrhage and capillary-leakage, which is part of the characteristics of a severe disease [8,14,15,16].

## 2. The Role of NS1 in Dengue Fever Diagnosis

Dengue fever (DF) is an arthropod-borne viral disease that is responsible for the greatest human disease burden [17,18]. The estimation of annual deaths is 10,000 and the estimation of apparent infection cases is between 100–400 million [17,19,20]. About 50% of the world’s population lives in dengue virus (DENV) endemic areas [20,21]. DENV infection is caused by four distinct serotypes (DENV 1-4), which are antigenically distinct [19]. All four serotypes may cause different forms of the disease, ranging from an asymptomatic infection and febrile illness to the more severe dengue hemorrhagic fever (DHF), dengue shock syndrome (DSS), and their related complications [22]. Infection with one of the DENV serotypes provides lifelong immunity to that specific serotype. However, immunity against other serotypes is limited and usually does not last for more than a few months [23]. Clinical diagnosis of DF as primary or secondary infection is important, as the greatest risk for DHF/DSS is considered to be with a second DENV serotype exposure (while the risk for DHF with a third or fourth dengue infection is very low) [23,24,25,26]. 

### 2.1. NS1 in Serum Samples 

DF is diagnosed using several assays, which are all based on DENV markers in the host. These markers are antibodies against the envelope protein of DENV (IgM\IgG) and the RNA of the virus [27,28]. 

The serological assays, based on the detection of IgM and IgG antibodies against the envelope protein, have several advantages. They are available at a low cost and do not require high-end equipment [29]. However, a significant disadvantage of this method is its inability to diagnose the disease in the acute phase of infection since IgM levels in the serum become detectable only after 3–5 days from symptoms onset, and peak approximately two weeks after the onset of fever [25,29,30,31]. Another critical issue regarding serology-based diagnosis is the cross-reactivity among flaviviruses [32]. Flaviviruses share common antigenic epitopes; therefore, antibodies against these common epitopes may cause false positive results when using an IgM- or IgG-based assay [30,32,33,34]. This issue makes the diagnosis of specific flavivirus infection more complex and less accurate, especially in regions that are endemic for several flaviviruses. Thus, for early detection of acute DF, the current immunological assays are not sensitive or specific enough [35,36,37]. Of note, dengue antibodies might be falsely detected due to past infection, infection with other flaviviruses, or past flavivirus vaccination [38], thus, making the diagnosis of secondary DENV infection challenging. Secondary DENV infection is considered a risk factor for severe dengue, emphasizing the clinical significance of diagnosis [39]. 

Molecular methods are based on DENV RNA detection by real-time RT-PCR. This method allows the detection of the virus at an early stage of the infection [40]. The main disadvantages of the molecular methods are their costs and availability, a serious consideration in low-income countries where DENV is endemic. Additionally, the window for virus detection using PCR is relatively narrow, and its highest sensitivity is achieved within the beginning of the illness. RT-PCR sensitivity varies greatly between various studies [41,42].

NS1-based assays may overcome several of the previously mentioned problems. NS1 levels in serum are highly correlated with DENV viremia [43] and become detectable at the beginning of the febrile illness, peaking 3 to 5 days following symptom onset. Furthermore, NS1 can still be detected in some cases until day 12 post-symptom onset, when RT-PCR can no longer detect viral RNA in most cases [38,44].

The commercial kits for detecting NS1 proteins are primarily based on ELISA. However, rapid one-step assays based on in-vitro immunochromatography are also available. Thus, NS1-based assays are readily available, simple to use as a rapid test, and cost-effective.

The assay’s sensitivity depends on the sampling time, the serotype of DENV, and the detection kit. Studies have evaluated the sensitivity of these assays and found it to be highest during the first days from fever onset (Table 1). A study conducted on returning travelers showed a sensitivity of 87% during the first three days of symptoms, that decreased to 69.4% when sampled during days 0–12 [38]. Other studies using a different commercial kit showed sensitivities of up to 95% during the first days [41,45,46]. The sensitivity of the NS1 antigen-capture assay was significantly higher for DENV-1 than for the three other serotypes [46,47].

The specificity of these assays was found to vary between 86% to 100% [9,38,41,53,54].

Sensitivity is significantly enhanced by combining the NS1 antigen kit with an IgM-based detection assay by approximately 28% [47,55]. The sensitivity of this combination is somewhat lower than the combination of RT-PCR together with an IgM-detection-based assay, and higher than the sensitivity of RT-PCR alone [47].

Several studies have found that during a secondary DENV infection, when IgG antibodies against NS1 are already present in serum, there is a lower sensitivity of the NS1-based assay [38,47,48]. Other studies have shown no significant difference in the sensitivity for the NS1-based assay between the first and second infection [56,57,58]. One study has shown a substantial decrease in the sensitivity (91.6% to 48.3%) of the NS1-based assay due to the presence of IgM antibodies against NS1 in the serum [58]

Another method that can contribute to detecting DENV is the Dried Blood Spot (DBS), a technique used when venous blood cannot be taken or where a laboratory does not exist. It requires a small amount of blood that can be collected from finger prick samples. This method involves minor discomfort, can be performed without medical assistance and does not require low temperature storage as is the case with serum samples. Several studies have shown the advantages of the DBS method [59,60,61] and its high sensitivity for NS1 detection, 98.7% during days 2–4 after the onset of the disease and 92.3% during days 5–7 [62].

The half-life of NS1 in plasma is about one hour [16]. The amount of free NS1 in the serum correlates with the viremia and, thus, it is a potential marker of the clinical outcome of DHF and DSS [47,63,64], suggesting that early quantitative detection of NS1 may provide a significant benefit. It can enable adequate follow-up for patients at high risk of developing DSS and DHF, whereas it can save unnecessary medications and hospitalization for patients with low risk. However, other studies found no correlation between NS1 levels in serum and disease severity [55,63].

In conclusion, NS1 is an important marker for detecting acute DENV infection. It offers an excellent serologic alternative to dengue viral RNA detection by RT-PCR, which is very sensitive and specific but expensive and not readily available in endemic regions (Figure 1). NS1 detection is useful for early DF diagnosis and whether quantifying the level of NS1 may identify patients at higher risk for severe dengue infection needs further studies.

### 2.2. Dengue NS1 Detection in Other Body Fluids

DENV-RNA and NS1 were also detected in urine and saliva samples [10,51,65,66,67] (Table 2). A study in Helsinki tested 14 Finnish travelers with DF, examining the presence of DENV-RNA and NS1 in saliva, urine and serum concurrently, and found that DENV-RNA in urine samples can be detectable at a later stage than serum samples [10] (Schemes Ι). This phenomenon, corroborated in other studies, is fundamental because it extends the time that DENV can be detectable using a molecular technique to the convalescent phase of infection. DENV-RNA levels in saliva were found to be similar to those in serum [10]. However, other studies found DENV-RNA levels in saliva were less sensitive, but taking longer to be detected in comparison to the levels in serum [65]. These results highlight the different kinetics of viral RNA in the urine and saliva.

NS1 levels in urine were found to correlate with reduced platelet counts and urine total protein concentration. Both urine NS1 levels and total protein concentration were found to correlate with serum DENV-RNA levels, suggesting that NS1 is the main DENV component in urine during the peak of the viremia [10].

A study performed in Japan [52] on 96 Japanese travelers with DF examined the utility of NS1 antigen detection in urine and serum samples. This study and others showed that, in some cases, NS1 could be detected in urine samples even after it reaches undetectable levels in serum. The NS1 detection rate in the serum showed a decrease after day 10, while the NS1 detection rate in the urine was relatively low (approximately 30%) but consistent until day 14. Similarly, detection based on the presence of urine DENV-RNA compared to the serum showed similar detection rates as those of NS1, with better results in detecting DENV-RNA in the urine than in the serum after day 11 [52,65].

A different study compared serum and urine samples obtained from 55 patients, of whom 19 had DF and 36 developed DHF with evidence of plasma leakage [9]. While NS1 levels in serum did not correlate with disease severity, the detection rates of NS1 in urine samples were higher in DHF patients, suggesting that the presence of NS1 in urine could be due to plasma leakage or the production of viral antigens by infected kidney cells [51]. In addition, the assays used in all the studies are designed and calibrated for serum; therefore, their sensitivities for urine and saliva must be evaluated. As for NS1 detection by ELISA in urine, most studies show high specificity but much lower sensitivity, around 100% and 20–68%, respectively [9,10,50,51,52]. Thus, more studies are needed to assess the performance and application of DENV diagnosis in urine and saliva.

### 2.3. CSF NS1 for Diagnosis of Dengue-Associated Encephalitis

Unlike other arboviral infections, DENV does not usually cause neurological manifestations [68]. The prevalence of CNS involvement in patients with dengue infection varies with the severity of dengue disease. However, the presence of CNS involvement does not affect the prognosis of dengue infection.

Although some studies suggest that DENV actively enters the CNS, this has not been demonstrated. Thus, it is still possible that the virus passively crosses the blood–brain barrier [69].

The presence of NS1 in CSF has not been well studied. One study examined the performance of IgM- and NS1-based assays (Pan-E Dengue Early ELISA) on CSF samples from dengue-infected patients vs. non-dengue patients obtained during autopsies [12]. The results showed that when used during viremia, the NS1-based assay can also detect NS1 in CSF with a sensitivity of 50%, while the sensitivity of the same kit in serum ranges from 60.4% to 91.6% [70]. As in serum, the combination of NS1 and IgM-based assays in CSF increases dengue diagnosis. However, similar to the observation in the serum, once IgM is detected, the sensitivity of NS1 Ag detection decreases significantly [70].

Another study from Brazil examined CSF samples from 209 patients with suspected viral meningitis or meningoencephalitis (26 of them died from suspected meningitis). All the samples were analyzed using RT-PCR, ELISA for NS1 (Pan-E Dengue Early ELISA), IgM, and a rapid immunochromatography test for IgG. Results demonstrate that out of 209 samples, 8 (3.8%) had shown positive results for DENV in at least one of the assays. Those findings are similar to other studies from Jamaica [71] and Vietnam [72].

Although more research is needed, these studies [12,73] show that NS1 kits for detecting NS1 antigen in serum can be used for CSF samples in dengue patients with neurological manifestations. As mentioned, it is recommended to use the combination of IgM- and NS1-based assays to increase the sensitivity of the diagnosis.

### 2.4. Detection of Dengue NS1 Is Tissue Specimens

To detect DENV in tissues, most studies have used RT-PCR, in situ hybridization, or immunostaining [74,75,76,77]. Another understudied alternative method for DENV tissue detection is the identification of NS1. The main application of this method is to reveal information about the cause of death that may lead to a better understanding of viral tropism in the human body in fatal cases.

Evidence for the presence of DENV antigens has been previously demonstrated in liver cells, brain inflammatory cells, spleen, lung, and pulmonary endothelium cells [59,60,62,74,77,78,79,80,81].

A study analyzed tissues from 13 children who died of DHF/DSS using immunohistochemistry and found NS1 in at least one tissue in all 13 cases using frozen and paraffin-embedded tissues [82]. The analyzed tissues were from organs, including the liver, spleen, lymph nodes, heart, lungs, kidney, bone marrow, and brain. In the liver, NS1 was identified in the cytoplasm of hepatocytes and Kupffer cells. Still, there was no evidence for NS1 in the sinusoidal endothelial cells. In the spleen, NS1 was identified in several mononuclear cells scattered within white and red pulps. In the lymph nodes, NS1 was found in the germinal centers. No evidence for NS1, or other DENV proteins, was found in other tissue specimens from the heart, lungs, kidneys, and bone marrow. However, other morphological changes related to DENV infection have been observed in some of these tissues.

Another study that assessed the usefulness of NS1-based assays on tissue specimens took 74 tissue samples from various organs from 23 fatal dengue cases [74]. For DENV detection, three different kits were used—Early ELISA (Panbio), Platelia NS1 (Biorad), and NS1 Ag Strip (Biorad). Per specimen, NS1 Ag Strip had the highest sensitivity of 78.3%; the Platelia NS1 sensitivity was 45.9%, and the Early ELISA sensitivity was 22.9%. Per case, the detection rates were 91.3%, 60.8%, and 34.7%, respectively. However, the overall sensitivity of this new approach in confirming the fatal cases was 87.0%. The specificity was 100% for all kits, and no cross-reactivity was observed with control tissues from fatal cases of yellow fever. It is important to note that each of the tissues had been detected in at least one case with one kit. For example, the NS1 Ag Strip, which had the best sensitivity, detected NS1 in the following tissue specimens: liver (91.3%), lung (71.4%), kidney (100%), brain (80%), spleen (66.6%) and thymus (100%). Similar results in liver and spleen tissue had been shown in another recent study [75].

The identity of the cells and tissues which DENV infects and replicates in humans are still incompletely understood. One possible explanation for the detection of NS1 in some of the tissue specimens is its presence in the blood supporting these tissues and, indeed, some of those organs that were well-diagnosed have high blood circulation. Yet, the liver seems to be an adequate tissue for NS1 detection.

These results encourage further studies and evaluation of the use of DENV NS1 in detection in tissues. Different postmortem preservation techniques and time of detection after death should also be examined. These methods are rapid, require less laboratory expertise and equipment, and cost less than the molecular and immunohistochemical methods that are currently in use.

This method, which can be performed following a needle biopsy, can dramatically improve the diagnosis and confirmation of postmortem fatal dengue cases in some parts of the world.

## 3. NS1 in Zika Infection

The Zika virus outbreak spread in the last years to a total of 89 countries and territories [83,84]. The Americas, and South Asia have significantly complicated flavivirus diagnosis. Cross-reactivity between DENV and ZIKV antibodies resulted in low specificity of ELISAs-based serology detecting flaviviral envelope-epitopes. More importantly, this cross-reactivity also occurs in viral neutralization assays, considered the gold standard for flavivirus diagnosis [85,86]. However, due to the substantial amino acid differences in NS1 between ZIKV and DENV, several ELISA kits have been developed to detect ZIKV NS1 antibodies [87,88,89,90] rather than envelope antibodies. In contrast to envelope-based assays, ZIKV NS1-based ELISA is highly specific and showed almost no cross-reactivity was detected in patients with previous DENV, WNV, YF, and Japanese encephalitis viral infections [89]. Interestingly, while IgG sensitivity was high, IgM sensitivity of the assay was demonstrated to be dependent on flavivirus background, and samples obtained from people living in areas with endemic ZIKV exhibited lower IgM sensitivity [89]. A study examining the sensitivity and kinetics of ZIKV NS1-based ELISA among travelers with ZIKV infection demonstrated higher IgM sensitivity of 79% in the first month and 68% up to two months post-symptoms onset [88] suggesting that levels of IgM against the NS1 may be declining much faster than IgM antibodies against the envelope protein. Indeed, while envelope IgM persisted for years after infection [91], NS1 IgM levels were shown to have a 75% decline by 44 days after symptom onset [92]. This suggests that measuring IgM against NS1 may be better suited for detecting the acute phase of the disease as it allows distinguishing between recent and past infections. Moreover, the limited cross-reactivity of NS1-based ELISA and its ability to serve as a first-line screening tool for travelers [93] allows the exclusion of ZIKV infection in travelers returning from ZIKV-endemic countries and, together with RT-PCR, minimizes the risk of false-negative results which is especially important for those who are pregnant or wish for preconception screening.

As a result of the excellent performance of the ZIKV NS1-based ELISA, DENV NS1-based ELISAs are currently being developed. They may provide better diagnostic capabilities for DENV, especially in areas with co-circulation of both DENV and ZIKV.

It is important to note that while NS1-based ELISA are a standard diagnostic tool in Europe and the Middle East, the Center for Disease Control and Prevention (CDC) uses an FDA-approved envelope-based ELISA and does not recommend serology testing for asymptomatic pregnant women or preconception screening [94].

## 4. The Role of NS1 for Other Flaviviruses Diagnosis

The development of assays for detecting DENV NS1 antigens in serum allowed easy and affordable diagnosis of dengue patients at the acute phase, as discussed above [38]. However, NS1 antigen detection tests against other flaviviruses are not commercially available. Two studies generated antibodies against WNV NS1 protein and investigated the secretion of WNV NS1 during WNV infection in vivo and in vitro using an ELISA assay [95,96]. NS1 was detectable up to day five in infected hamsters [95] and up to day three post-infection in mice [96] suggesting that NS1 secretion during infection is low. Still, the detection of WNV NS1 is comparable to the detection of WNV RNA by qRT-PCR.

## 5. The Role of Dengue NS1 Testing to Differentiate Dengue vs. COVID-19 Infection

The pandemic of SARS-CoV-2 rapidly spread around the globe, including areas endemic to dengue. DF and COVID-19 are difficult to distinguish as they share similar clinical and laboratory features [97]. Interestingly, recent reports from several countries in Asia showed cross-reactivity of either IgM and/or IgG antibodies between these two, despite being unrelated viruses [98,99,100,101]. This effect might have an impact on the clinical outcome if diagnosis is delayed. On the other hand, other evidence downsized the concern of cross-reactivity [102]. Hence, further studies are warranted to explore the pathophysiology of this cross-reactivity and to better elucidate the accuracy of the different commercial kits. Given that a possible cross-reactivity between DENV and SARS-CoV-2 could lead to false-positive results for both diseases and the consideration that DENV has been extensively spread in recent decades, a serological test alone might be misleading. However, using dengue NS1 antigen for diagnosis can prevent cross-reactive results and leads to a more accurate diagnosis.

## 6. In Summary

NS1 is an essential flavivirus protein that has a central role in efficient viral RNA replication. This protein accumulates intracellularly and is secreted from infected cells. Soluble NS1 is present in blood serum and other body fluids and, therefore, can be used as an antigen-detection diagnosis. Since antibodies are produced against this protein they can be used for serology diagnosis. There are advantages of using NS1-based diagnosis (antigen and/or antibodies) in several aspects:

Epidemiologically, its highly specific properties differ in each flavivirus, and therefore minimize cross-reactive results among the flaviviruses group. The amounts of NS1 antigen are high during dengue infection, while during Zika infection, the amount of NS1 antibodies are high. Better methods of detecting low concentration of NS1 may also allow its use in the diagnosis of other flaviviruses as well. Currently a reliable serological diagnosis to differentiate each virus within the family of flaviviruses is less than optimal and introducing NS1-based serology can give more precise diagnosis and better epidemiological understating of the global distribution of each flavivirus.

From the clinical point of view, a correct identification of the virus might have great significance. For example, Zika virus infection is highly important in a pregnant woman as there is a need for further evaluation of in-utero infection.

Regarding DF, the amount of NS1 antigen is important as it correlates with the level of viremia, which makes it a potential marker of severe disease.

Regarding a virology aspect, detection of the specific NS1 antibodies against each DENV serotype can give an assessment of previous infection with DF and which serotype it was.

Regardless, NS1-based diagnosis does not replace other serology, especially when the diagnosis is made long after the infection, nor the very sensitive but expensive PCR.

## Figures and Tables

**Figure 1 viruses-15-00572-f001:**
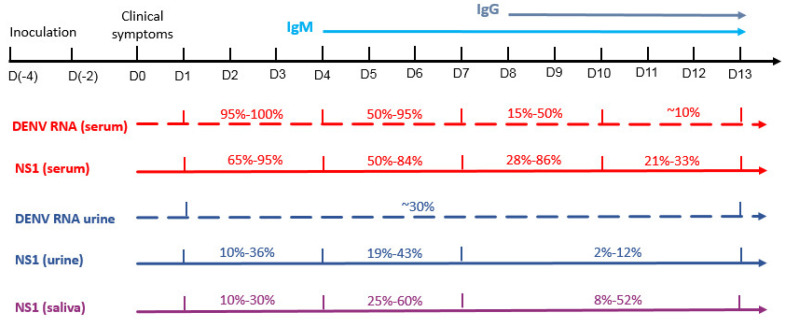
Schematic time line (D = days) of the different DENV detection methods and their sensitivities. The sensitivity range reflects the variety of results in different studies.

**Table 1 viruses-15-00572-t001:** Comparison of sensitivity and specificity of NS1 DENV detection in serum samples, accordance to timeline from symptoms-onset.

Source	Method	% Sensitivity According to Days Post Symptoms Onset													Specificity %
		1	2	3	4	5	6	7	8	9	10	11	12	13	
Blessmann et al., 2020 [48]	RDT (SD Bioline Dengue Duo)	78.3			60	42.9	28.6			ND	ND	ND	ND	ND	>96
	ELISA (Platelia, Bio-Rad)	65.2			50	42.9	28.6			ND	ND	ND	ND	ND	>96
Liu et al., 2020 [46]	RDT (Bio-Rad)	89.9			80.6			ND	ND	ND	ND	ND	ND	ND	94.6
	RDT (SD Dengue Duo)	95.7			83.6			ND	ND	ND	ND	ND	ND	ND	91.9
	RDT (CTK)	95.7			82.1			ND	ND	ND	ND	ND	ND	ND	73
Hunsperger et al., 2016 [49]	ELISA (InBios)	93	93	85	83	84	60	57	47	86	67	ND	ND	ND	ND
	ELISA (Panbio)	86	93	78	71	71	52	50	33	57	33	ND	ND	ND	ND
Andries et al., 2016 [50]	ELISA (NS1)	94.4			66.7				21.3					ND	ND
Andries et al., 2015 [51]	ELISA (NS1)	83.3	91.6		79		55.9		29.1						ND
Fuchs et al., 2014 [38]	ELISA (Panbio)	87.5			ND	ND	ND	ND	ND	ND	ND	ND	ND	ND	88.8–100
		73.9					ND	ND	ND	ND	ND	ND	ND	ND	
		72.5									ND	ND	ND	ND	
		69.4												ND	
Saito et al., 2015 [52]	ELISA (Platelia, Bio-Rad)	>90										33			ND
Huhtamo et al., 2010 [41]	ELISA (Platelia, Bio-Rad)	78.6			74.1		84.2		70.6			ND	ND	ND	100
Ramirez et al., 2009 [53]	ELISA (Panbio)	60.9							ND	ND	ND	ND	ND	ND	94.4
	ELISA (Platelia, Bio-Rad)	71.3							ND	ND	ND	ND	ND	ND	86.1
	RDT (AG Strip (BIO-RAD)	67.8							ND	ND	ND	ND	ND	ND	94.4
Chuansumrit et al., 2011 [9]	ELISA (Platelia, Bio-Rad)	ND	ND	94.7								ND	ND	ND	100
	RDT (Bio-Rad)	ND	ND	89.5								ND	ND	ND	100

Abbreviation: ND—Not done; RDT—Rapid diagnostic test; ELISA—Enzyme-linked immunosorbent assay.

**Table 2 viruses-15-00572-t002:** Comparison of sensitivity and specificity of NS1 detection in urine and saliva samples, accordance to timeline from symptoms-onset.

Source	Method	% Sensitivity According to Days Post Symptoms Onset													Specificity %
		1	2	3	4	5	6	7	8	9	10	11	12	13	
** *Urine* **															
Andries et al., 2016 [50]	ELISA	9.9			19.2				3.4					ND	100
	RDT	11.1			21				2.3					ND	ND
Andries et al., 2015 [51]	ELISA	0	10.8		28.6		18.6		0						ND
Saito et al., 2015 [52]	ELISA	36					43						12		ND
	real-time RT-PCR (for reference)	32					30						33		ND
Korhonen et al., 2014 [10]	ELISA	ND	59.1					50							100
Chuansumrit et al., 2011 [9]	ELISA	ND	ND	68.4								ND	ND	ND	100
	RDT	ND	ND	52.6								ND	ND	ND	100
** *Saliva* **															
Andries et al., 2016 [50]	ELISA (NS1-Saliva)	25.5			29.9				8.3					ND	ND
	RDT (NS1-Saliva)	14.9			25.7				10.2					ND	ND
Andries et al., 2015 [51]	ELISA (NS1-Saliva)	16.7	33.7		41		28.8		10.9						ND
Korhonen et al., 2014 [10]	Antigen EIA (NS1-Saliva)	ND	59.1					52.2							95

Abbreviation: ND—Not done; RDT—Rapid diagnostic test; ELISA—Enzyme-linked immunosorbent assay.

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
