# Peer review of "The Role of NS1 Protein in the Diagnosis of Flavivirus Infections"

_viruses, 2023, doi:10.3390/v15020572_

Round 1

Reviewer 1 Report

In this review, author described “The role of NS1 protein in the diagnosis of flavivirus infections”. DF is diagnosed using several assays, which are all based on DENV markers in the host. These markers are antibodies against the envelope protein of DENV (IgM\IgG) and the RNA of the virus. Both assays have some limitation. NS1-based assays may overcome these limitations. The higher concentration of NS1 directly correlates with disease severity and increased viremia. NS1 is secreted from infected cells within hours after viral infection, and thus immunodetection of NS1 can be used for early serum diagnosis of dengue fever infections instead Real-Time (RT)-PCR. This method is fast, simple, and affordable, and its availability could provide an easy point-of-care testing solution for developing countries.  It is clearly a big effort of author for doing this study, but despite of positive outcomes I have some suggestion where it can be improved.

Comments

1.     The whole review is describing about role of NS1 protein in DENV infection, but title of review is the role of NS1 protein in the diagnosis of flavivirus infections. I think author can describe more about NS1 protein in case of other flaviviruses or author can change the title.

2.     In line 65, 78, 139, 185 and 207. Author described the role of NS1 in Dengue Fever diagnosis in line 65. In line 78,139, 185 and 207 all are parts of “The role of NS1 in Dengue Fever diagnosis”. If author can add a, b, c and d in front of line 78,139,187 and 207, that will be clearer to the readers.

3.     In line 177-179, Schemes 1 is blurry, it’s not clearly visible. Please make it clear

4.     In line 50, please put a space between replication and [1] instead of replication[1]

5.     In line 52 and in most of the line author did not give a space before the reference. Please give space before the reference.

Reviewer 2 Report

Dear authors, please addresse the following recommendations:

Major(s)

1.      Please revised the entire manuscript and add the respective missing references to many sentences along the manuscript describing previously published scientific findings.

2.      The review will benefit from including a figure of what the dynamics for antibodies responses, virus, and NS1 are during well-known flavivirus infections such as dengue, Zika or yellow fever.

3.      Please clarify and better justify the introduction of this last section called “The role of Dengue NS1 testing to differentiate dengue vs. COVID-19 infection” in the manuscript. Although few studies have described the existence of potential cross-reactivity between DENV and SARS-COv-2, this is still highly controversial.  

4.      In addition to a summary, authors should conclude why this revision on NS1 diagnostic tools and flaviviruses is important, pro- and cons- of using one or another assay either commercial or non-commercial, to differentiate flavivirus infections in endemic areas, potential prognosis biomarker, and future perspectives, etc.

Minor(s)

1.      In line 23, authors say “Nonstructural protein 1 (NS1) is a glycoprotein encoded by most viruses from the flavivirus genus”. Do the authors have knowledge of any flavivirus that does not encode for NS1 in their genome? I would suggest changing it for “Nonstructural protein 1 (NS1) is a glycoprotein among flavivirus genus” or something related.

2.      Line 45 and 46, Dengue, Yellow fever, should be written in lowercase.

3.      Line 65, in the title “Dengue fever diagnosis”, Dengue should be written in lowercase. Please correct these names along the entire manuscript.

4.      Once introduced, abbreviations must be used along the entire manuscript, for instance, DENV, dengue virus, Dengue virus.

Reviewer 3 Report

In this review Fisher et al. describe the current knowledge, rationale and advantages of using dengue NS1 based assays as a more relevant diagnostic tool. Flavivirus NS1 is an early marker of flavivirus infections. It plays an important role in viral pathogenesis and host immune response, and acts as an antigen to elicit specific antibodies in the host.  Dengue fever is a major mosquito-borne viral infection in many parts of the world, and a review on current understanding of better, more efficient ways of early and specific detection of acute flavivirus infections is a welcome and important contribution.

Comments-

·       Ln 37: rephrase “should be used” in sentence. Would the NS1 based assay be able to differentiate between multiple flaviviruses from a single serogroup?

·       Ln 38: Consider including ‘Dengue virus’ as the main focus of the review is primarily on dengue virus.

·       Ln 41 (Keywords): Dengue virus instead of Dengue, Zika virus instead of Zika

·       Ln 48: To maintain consistency, please change “C-prM-E” to “C, prM and E”, and define C, prM and E

·       Ln 53: “….postranslationally modified in the lumen of the endoplasmic reticulum” – Please add reference

·       Ln 56: “…in serum and other body fluids” – Please add  reference

·       Ln 63, 92, 116, 145-146: please check/correct sentence structure - “All contributing to possibility…”; “Making even the diagnosis…”; “However, higher than...”; “…and less sensitive and longer to be detected..”

·       Ln 66: Please support this statement with appropriate reference or rephrase “the world’s most common” part. Dengue causes a significant disease burden globally but it isn’t geographically as widespread as, say, WNV so it isn’t clear how DF is the world’s most common arbovirus disease. Also consider including a more recent reference (perhaps, Messina et al 2019? https://pubmed.ncbi.nlm.nih.gov/31182801/)

·       Ln 109, and Table 1: What is the reason for the four rows corresponding to Fuchs et al? For days 10-12, the Table does not show 70% sensitivity. Similarly, please check whether reference 34 is correct for 95% sensitivity mentioned in text (Ln 111) and matches with values in Table 1, and whether both references 39 & 40 support the claim for DENV1 (Ln 112).

·       Ln 137: “…may identify patients with high risk for severe dengue infections”. Since in the previous paragraph, there is mention of studies both supporting a correlation and showing no correlation of NS1 serum levels and severe disease, it would be helpful to add the basis on which this conclusion was deduced.

·       Ln 164-165: “…most studies show high specificity, but much lower sensitivity..”. Table 2 shows specificity for only one study that is referenced in text and the others as ‘Not Done’. Please double check sentence, references and Table entries for accuracy.

·       Ln 287: change “antibodies cross-reactivity of either IgM and or IgG between these two” to ‘cross-reactivity of either IgM and/or IgG antibodies between these two’. Please check for accuracy.

·       Ln 290: change “spread in the last decades” to ‘spread in recent decades’.

·       Ln 291: “and thus must currently be the role of the diagnosis”. Meaning is not clear, consider re-phrasing.

Round 2

Reviewer 2 Report

Dear editor.

The authors have addressed all the reviewer comments.